# Changes in Neuropeptide Prohormone Genes among *Cetartiodactyla* Livestock and Wild Species Associated with Evolution and Domestication

**DOI:** 10.3390/vetsci9050247

**Published:** 2022-05-23

**Authors:** Bruce R. Southey, Sandra L. Rodriguez-Zas

**Affiliations:** 1Department of Animal Sciences, University of Illinois at Urbana-Champaign, Urbana, IL 61801, USA; rodrgzzs@illinois.edu; 2Department of Statistics, University of Illinois at Urbana-Champaign, Urbana, IL 61801, USA

**Keywords:** neuropeptide, prohormone, domestication, *Cetartiodactyla*, evolution

## Abstract

The impact of evolution and domestication processes on the sequences of neuropeptide prohormone genes that participate in cell–cell signaling influences multiple biological process that involve neuropeptide signaling. This information is important to understand the physiological differences between *Cetartiodactyla* domesticated species such as cow, pig, and llama and wild species such as hippopotamus, giraffes, and whales. Systematic analysis of changes associated with evolutionary and domestication forces in neuropeptide prohormone protein sequences that are processed into neuropeptides was undertaken. The genomes from 118 *Cetartiodactyla* genomes representing 22 families were mined for 98 neuropeptide prohormone genes. Compared to other *Cetartiodactyla* suborders, *Ruminantia* preserved PYY2 and lost RLN1. Changes in GNRH2, IAPP, INSL6, POMC, PRLH, and TAC4 protein sequences could result in the loss of some bioactive neuropeptides in some families. An evolutionary model suggested that most neuropeptide prohormone genes disfavor sequence changes that incorporate large and hydrophobic amino acids. A compelling finding was that differences between domestic and wild species are associated with the molecular system underlying ‘fight or flight’ responses. Overall, the results demonstrate the importance of simultaneously comparing the neuropeptide prohormone gene complement from close and distant-related species. These findings broaden the foundation for empirical studies about the function of the neuropeptidome associated with health, behavior, and food production.

## 1. Introduction

Domestication typically encompasses the separation of a species from its natural habitat and physiological and behavioral modifications through artificial selection and environments [1]. This interplay is uniquely demonstrated in the mammalian superorder *Cetartiodactyla* that includes domesticated livestock species as well as species adapted to diverse environmental conditions including alpine, desert, and marine environments, and management practices including precision management [2,3,4]. Out of the four *Cetartiodactyla* suborders, three suborders have at least one domesticated species: *Ruminantia* (sheep, cattle, goats, and deer), *Suina* (pigs and peccaries), and *Tylopoda* (camels, llamas, and alpaca). These domesticated terrestrial *Cetartiodactyla* species are managed for a wide variety of purposes including food, fiber, companion, show, and draught purposes. *Whippomorpha* (hippopotamuses and whales), the fourth *Cetartiodactyla* suborder, encompasses species abundant in the wild, and also present in zoos and aquariums participating in conservation and scientific studies [5].

Livestock domestication started approximately 11,000 years ago with domestic cattle (*Bos taurus*), domestic goats (*Capra hircus*), domestic sheep (*Ovis aries*), and domestic pigs (*Sus scrofa*) [2,3,4]. *Tylopoda* were domesticated 3000 to 7000 years ago [6] and reindeer (*Rangifer tarandus*) are partly domesticated with both wild and domesticated herds [7]. Domestication involves the modification of biological processes associated with the regulation of behaviors such as aggression, fear, and flight. Domestication also elicits changes in biological processes associated with fertility, metabolism, development and growth, feeding and reproductive behavior, and health through artificial selection, breeding, and management [1,2,3,4].

Artificial selection has led to population genetic changes that, in turn, elicited behavioral and physiological changes in the domesticated populations. Underlying these biological changes are processes associated with cell-to-cell signaling that is characterized by small neuropeptides conveying chemical messages between neurons and other cells when docking on prohormone receptors. Aggression, bonding, and stress response are modulated by the neuropeptides adrenocorticotropic hormone (produced from proopiomelanocortin; POMC), oxytocin (produced from oxytocin/neurophysin I prepropeptide; OXT), arginine vasopressin (produced from arginine vasopressin; AVP), and corticoliberin (produced from corticotropin releasing hormone; CRH) [8]. Appetite and feeding behaviors are modulated by neuropeptide insulinlike growth factor 2 (IGF2), neuropeptide Y (NPY), and ghrelin (produced from ghrelin and obestatin prepropeptide; GHRL).

The evolutionary processes of natural selection mutation, gene flow, and genetic drift have also resulted in changes in neuropeptide sequences and associated biological processes. Sequence comparison of the genes coding for the neuropeptides’ glucagon (GCG) and glucose-dependent insulinotropic polypeptide (GIP) across species offers insights into the impact of evolution on associated processes [9]. The mutation of the melanocortin receptor 1 (MC1R), a receptor for POMC neuropeptides, is an important component of coat color diversity in wild and domestic species [10]. Moreover, while artificial and natural selection processes can coexist, different selection pressures between wild and domestic livestock species have been reported [11]. Analysis of the melanocortin system showed that MC1R had the largest coevolutionary influence, while the POMC and agouti-related neuropeptide (AGRP) had the lowest and third lowest coevolutionary influence, respectively [12].

The impact of domestication on neuropeptides cannot be predicted directly from the species genome or transcriptome because neuropeptides are the result of complex processing of a larger protein sequence. These proteins are encoded by neuropeptide prohormone genes that contain a signal peptide which directs the protein for secretion. After removal of the signal peptide [13], the resulting protein sequence undergoes cleavage at sites compatible with the furin cleavage motif [14]. This motif consists of the combination of an arginine or lysine at the cleavage site with an adjacent or near-adjacent position containing arginine or lysine. Subsequently, cleaved peptides may undergo additional post-translation modifications to form bioactive neuropeptides.

A bioinformatic pipeline that works with protein sequences and accommodates the complex processing of neuropeptide prohormone gene products has been developed [15,16]. This bioinformatics strategy succeeded in the identification and characterization of neuropeptide prohormone genes that have been lost or undergone the duplication part of the evolutionary process [17,18,19]. The objective of the present study was to characterize the changes in neuropeptide prohormone genes associated with domestication and evolutionary processes. To accomplish this objective, a rich database of wild and domesticated *Cetartiodactyla* species was assembled. The resulting database offered a unique opportunity to identify neuropeptide prohormone gene gain and loss, and hybridization events that occurred along the processes of domestication and evolution [20,21,22].

## 2. Materials and Methods

The complete genomic assemblies of 118 *Cetartiodactyla* species distributed across all 4 suborders and encompassing 22 families were downloaded from the National Center for Biotechnology Information (NCBI) repository [23]. Table 1 summarizes the species studied and Appendix A includes lists of all species and their corresponding identifiers. The *Ruminantia* suborder was represented by 82 ruminant species including cattle, deer, goats, and sheep distributed across 6 families. The *Suina* suborder was represented by 2 species including pigs and peccaries. The *Tylopoda* suborder was represented by 7 species from 2 tribes including camels and llamas. The *Whippomorpha* suborder was represented by 27 species including whales and hippopotamus.

Different groupings of the *Cetartiodactyla* species were identified based on domestication status, suborder, family, and subfamily. Individual species were classified as domesticated (12 species) or wild (106 species). The wild Bactrian camel, *Camelus ferus*, was excluded from the previous classification due to uncertainty on the domestication status of the sequenced individuals [24]. The wild species from the suborders *Ruminantia*, *Suina*, and *Tylopoda* were grouped as terrestrial (non*Cetacea*) species (91 species), while *Cetacea* species (*Whippomorpha* species excluding the hippopotamus, *Hippopotamus amphibious*) are aquatic. *Ruminantia* species were also divided into domestic (8 species) and wild (74 species).

A catalog of neuropeptide gene products (prohormone proteins) was assembled and the predicted protein sequences for each species were obtained following the bioinformatic pipeline [15,16]. Since *Cetartiodactyla* species were unannotated, additional steps within the pipeline were taken to identify the neuropeptide prohormone genes. The longest prohormone protein isoforms were compiled from a curated list of 98 mammalian prohormone protein sequences identified in *Cetartiodactyla* [15,17,18,25,26,27]. The most probable location for each neuropeptide prohormone gene was identified using TBLASTN [28] using default settings and with the low-complexity filtering disabled. An approximate 10,000 bp region surrounding the most probable gene location was extracted and the corresponding protein sequence was predicted from the extracted region using the gene prediction tool Genewise [29] with default settings. Individual sequence inspection ensured that all predicted sequences corresponded to the same longest known protein isoform. Variations in assembly depth and quality across the genomes could result in incomplete sequences or apparent sequence duplication. Incomplete, inconsistent, or potentially duplicate predictions were evaluated and additional searches using alternative locations, spanning intronic regions to accommodate genes that have short exons, and relaxing search criteria with and without composition-based statistics [30] were used to improve the prediction.

Following the compilation and validation of the neuropeptide gene protein sequences, the similarity between species was studied. The alignment of protein sequences across species by prohormone was implemented using the sequence aligner MAFFT [31]. Alignment accuracy was optimized using the L-INS-I parameterization that enables the alignment of sequences containing sequences flanking around one alignable domain [31]. Phylogenetic gene trees of protein sequences were computed for each neuropeptide prohormone gene using the software PhyML [32] with default parameters, and tree visualization used the software ASTRAL with default parameters [33,34,35,36]. The genetic distances between species trees and individual prohormone trees were estimated using Pearson correlation coefficients. The protein distances within prohormones and across the different groupings were estimated using the mean protein evolutionary distance approach [37].

The estimation of neuropeptide prohormone gene protein distances was ensued by the identification of the evolutionary/domestication paradigm best supported by the 118 species studied. An evolutionary model [38] accounting for information on amino acid aromaticity (A), composition (C), polarity (P), and side chain volume (V) [39,40,41] was fitted using the Jones-Taylor-Thornton (JTT) substitution model [42]. The final model best supported by the sequences and species studied was selected using Akaike information criterion [43].

## 3. Results

### 3.1. Prohormone Identification across Species

All 98 prohormone protein sequences were identified across all 118 *Cetartiodactyla* species but varied across taxonomic groups. A notable finding from the comparison across species was that relaxin 1 (RLN1) has been lost in *Ruminantia* but is present in all other *Cetartiodactyla* suborders. Furthermore, prohormone peptide YY2 (PYY2) was only reliably detected across the *Ruminantia* suborder including complete predictions in the *Giraffidae* and *Bovinae* subfamilies and a partial PYY2 prediction in the *Cervidae* subfamily. Protein sequences from AVP, secretin (SCT), tachykinin precursor 4 (TAC4), and VGF nerve growth factor inducible (VGF) were completely or partially identified in at least 20% of the *Cetartiodactyla* species. Apelin (APLN), CART pre-propeptide (CARTPT), NPY, and somatostatin (SST) protein sequences were found to be highly conserved.

Within species, the genome assemblies of argali (*Ovis ammon*), Alpine ibex (*Ammotragus lervia*), and blue whale (*Balaenoptera musculus*) enabled the prediction of 56%, 53%, and 79% of neuropeptide prohormone genes, respectively. The hartebeest (*Alcelaphus buselaphus*) genome supported the complete or partial prediction of 69% of the neuropeptide prohormone genes. When compared to closely related species, the challenge in predicting the totality of the prohormone protein sequence completely was primarily due to depth and quality variations along the individual assemblies rather than species differences. Limitations in sequence coverage, contig assembly including frame orientation, and splice site prediction prevented an accurate census of neuropeptide prohormone gene gains and losses despite the multiple specifications of the predictive algorithms used in the present study.

### 3.2. Species Tree Derived from Prohormone Sequences

#### 3.2.1. Species Tree

The species tree followed the expected species relationships between and within the four *Cetartiodactyla* suborders (Figure 1). The species tree excluded the RLN1 and PYY2 gene trees because these genes were suborder-specific. Among the *Whippomorpha* species, the hippopotamus was more distant to the rest of the species in the suborder that correspond to the infraorder *Cetacea* which in turn is distributed into the parvorders *Mysticeti* (baleen whales) and *Odontoceti* (toothed whales). The tree also demonstrates that the comparative analysis of neuropeptide protein sequences resulted in the expected organization of species into known families and subfamilies within the *Ruminantia* suborder.

#### 3.2.2. Correlation of InterSpecies Distances Based on Individual and All Neuropeptide Prohormone Genes

To understand the variation in species distances across neuropeptide prohormone genes, the distances estimated within the neuropeptide prohormone gene tree were correlated to the distances from the species gene tree. The correlation of interspecies distances between individual and the species gene trees averaged 0.77 and ranged from 0.25 to 0.92. The species distance for most individual neuropeptide prohormone genes (69 genes) was highly correlated (between 0.8 and 0.9) with the distance estimate from the species gene tree. Neuropeptide prohormone genes that are highly conserved tended to provide distance estimates less correlated with the estimates from the species gene tree, relatively with less conserved neuropeptide prohormone genes. This was reflected by the number of unique sequences and the length of the predicted protein sequence. The interspecies distance estimated from chromogranin A (CHGA), prodynorphin (PDYN), and the thyrotropin releasing hormone (TRH) were highly correlated (correlation >0.9) with the distance from the species gene tree. On the other hand, the interspecies distance estimated from AVP, insulin (INS), natriuretic peptide C (NPPC), prokineticin 2 (PROK2), parathyroid hormone (PTH), and SST had lower correlations (correlation <0.50) with the distance from the species gene tree resulting from extremely highly conversed protein sequences across species.

The distance between *Ruminantia* species computed from individual neuropeptide prohormone genes had higher correlations with the distance computed from the species gene tree relative to other species. This trend may reflect the higher proportion of ruminant species among all species used. A higher average distance between species was estimated in the second most represented suborder, *Whippomorpha*. The hippopotamus, sperm whale (*Physeter catodon*), and baiji (*Lipotes vexillifer*) species were more distant from other *Whippomorpha* species.

### 3.3. Evolutionary Model

The evolutionary model use accounts for information on amino acid aromaticity (A), composition (C), polarity (P), and side chain volume (V) to understand the impact of evolutionary and domestication changes within and across suborders. Table 2 summarizes the number of neuropeptide prohormone genes by evolutionary model specification. Most neuropeptide prohormone genes exhibited a nonzero estimate for at least one parameter; parameters A and V commonly had negative estimates, while parameter C typically had positive estimates, and P tended to have positive estimates (Table 2). This relationship between parameters is consistent with the correlations of individual amino acid coefficients for each parameter in the evolutionary model. Both the A and V parameters are positively correlated (~0.7) and negatively correlated with the C and P parameters (−0.35 to −0.47). The C and P parameters are positively correlated (0.4) which increased to 0.81 when cysteine is excluded.

The domestic group of species had a higher proportion of neuropeptide prohormone genes (40%) than the wild grouping with nonzero parameter estimates especially involving the C parameter. A smaller proportion of parameter changes occurred between domesticated and wild *Ruminantia* species. There were seven neuropeptide prohormone genes where the modified parameters only occurred in domestic (1) or wild terrestrial (6) groups. These neuropeptide prohormone genes included apelin (APLN), adenylate cyclase activating polypeptide 1 (ADCYAP1), arginine vasopressin (AVP), cholecystokinin (CCK), growth hormone releasing hormone (GHRH), torsin family 2 member A (salusin-containing isoform, TOR2X), and VGF nerve growth factor inducible (VGF), and the averages of the parameter changes for these neuropeptide prohormone genes are summarized in Table 3. The positive A parameter estimate for APLN implies a preference for aromatic amino acids with domestication. In the wild groupings, the A and V parameters were negative estimates and the C parameter had positive estimates, implying a preference against aromatic amino acids with domestication.

## 4. Discussion

### 4.1. Prohormone Complement

The integration of bioinformatics prohormone prediction, compilation, characterization, and analysis offered insights into changes of neuropeptide genes associated with evolutionary and domestication processes across *Cetartiodactyla* suborders and species. As expected of orders that encompass a wide range of species, the *Cetartiodactyla* assemblies differed in depth and quality across the genome. The *Cetartiodactyla* species tree was virtually identical to the expected tree [6,44,45]. The strategy of using evolutionary proximal species for gene prediction in weaker assemblies [46] enabled the recovery of sequences and the results indicated overall sequence consistency across taxonomic groups. The evolutionary proximal strategy minimized the identification of differences between species that could be a result of assembly limitations.

A remarkable finding was that *Suina* species contained the AUG initiation codon but all other *Cetartiodactyla* species in this study contained a nonAUG initiation codon of neuropeptide W (NPW) present in other mammalian species [47]. The absence of the AUG initiation codon can result in incorrect prediction of the start of the actual coding region of NPW. The unique feature of the neuropeptide NPW is particularly important because pigs are regularly used as biomedical models due to their higher genome similarity to humans than rodents and several health and behavioral processes are modulated by NPW.

Another finding from the bioinformatics pipeline centered on sequence variations in cortistatin (CORT). A 15-nucleotide insertion in signal peptide of CORT was detected in the domestic goat and does not impact the cortistatin neuropeptides. The potential evolutionary difference was detected in the assemblies of four domestic goat breeds currently available and this insertion was not present in other *Capra* or *Caprinae* species.

Comparisons of the predicted prohormone sequences highlighted and refined existing knowledge of the neuropeptide genes RLN1, PYY2, islet amyloid polypeptide (IAPP), and galaninlike peptide (GALP). Our predictions of RLN1 extended the loss of bovine RLN1 [17] to all *Ruminantia* species after the split from *Whippomorpha*. Oppositely, PYY2 was detected solely in *Ruminantia* species. Species from other suborders either lacked a complete prediction or had indeterminate matches (*Whippomorpha*) supporting that PYY2 is a pseudogene outside *Ruminantia* [48]. While GALP was detected in most species, all *Cetartiodactyla* suborders had different terminal regions and some *Bovidae* species lacked the initial initiation region. *Tylopoda* and *Cetacean* species presented IAPP sequences encompassing discrepancies consistent with pseudogenes including stop codons and lack of a signal peptide. Chacoan peccary (*Catagonus wagneri*) and all the camelid species studied, along with previous reports on the pig IAPP [18], lack the sequence corresponding to traditional cleavage sites, although analysis of RNA-seq data indicated that IAPP is expressed in pigs [49].

The comparative analysis of neuropeptide genes between *Cetartiodactyla* species identified sequence differences that could impact the levels of bioactive neuropeptides including TAC4, insulinlike 6 (INSL6), gonadotropin releasing hormone 2 (GNRH2), and prolactin releasing hormone (PRLH). Noticeably, TAC4 exhibited sequence differences among the *Cetartiodactyla* species studied, yet no sequence encompassed the cleavage motif reported in human and mouse [50]. Similarly, the start and terminal regions of INSL6 in pigs and camelids were different from the other *Cetartiodactyla* species that, in turn, were similar to the corresponding human and rodent sequences [51]. While the N-terminal cleavage site of the INSL6 A chain was conserved across species, the C-terminal cleavages sites of the INSL6 A and B chains remains uncharacterized among *Cetartiodactyla* species. Likewise, while the signal peptide for GNRH2 was six amino acids longer in all the species of the *Bovinae* subfamily, the expected cleavage motif for the gonadoliberin-2 peptide was missing from all the *Whippomorpha* species. Similarly prominent, all *Cetacea* species had an 8-amino acid deletion within the PRLH signal peptide that was not present in the *Hippopotamus amphibius* or other *Cetartiodactyla* species and this sequence difference resulted in a shorter predicted prolactin-releasing peptide 31 (PrRP31). On the other hand, a 10 bp-insert in all *Ruminantia* species results in a longer PRLH terminal region, albeit this region is not part of a known bioactive peptide.

The variation in POMC protein sequences across the *Cetartiodactyla* species could impact multiple neuropeptides. While the complete POMC sequence was predicted in *Tylopoda* species, the sequence lacked a cleavage site within the N-terminal peptide (NPP) necessary to produce the pro-γ-MSH, and in turn, the small amidated γ_1_-MSH. While no peptides associated in pro-γ-MSH region were detected by mass spectrometry [27], γ_1_-MSH has been characterized in human [52] and cattle [53]. The N-terminal cleavage site was detected in all the other *Cetartiodactyla* species studied. A similar phenomenon is observed among rodent species [54] with the cleavage site missing in *Muridae* yet observed in other *Rodentia* families. Remarkably, both *Tylopoda* and *Muridae* species include the sequence for the larger γ_3_-MSH peptide. The role of γ-MSH peptides in health and behavior processes encompasses sodium metabolism and blood pressure regulation [55,56,57] and the injection of γ_1_-MSH in the left ventral tegmental area of rats induced grooming [58].

Neuropeptides from calcitonin (CALC) genes have many different roles including the regulation of calcium, vasodilation, inflammation, migraine, pain, and hypothermia [59,60,61,62]. Related with domestication behaviors, an SNP in the calcitonin receptor-stimulating peptide (CRSP) gene cluster differentiated pure-breed (mating determined by human breeding practices) from free-breeding (mating is not artificially restricted) dogs [63]. Moreover, the calcitonin receptorlike receptor (CALCRL) was differentially expressed in the pituitary of tame and aggressive foxes [64]. In the present study, the identification of four calcitonin or CRSP genes across all *Cetartiodactyla* taxa was consistent with other mammals except primates and rodents [65]. While calcitonin-related genes were detected, the comparison of cleavage site locations indicated that the cleavage sites necessary to form all known neuropeptides such as a calcitonin gene-related peptide 1 are absent from both CRSP2 and CRSP3.

### 4.2. Evolutionary Model

The differences in the evolutionary model amino acid aromaticity (A), composition (C), polarity (P), and side chain volume (V) parameters enabled the identification of patterns of amino acid property changes in the prohormone sequences and thus neuropeptides across *Cetartiodactyla* taxa. The similarity in the number of changes between all species and all *Ruminantia* species was due to the high proportion of *Ruminantia* species and lower similarity was observed with family or subfamily groupings. The relatively high proportion of changes within *Tylopoda* was due to the limited number of species and the close relationship between *Tylopoda* species. Within taxonomic groupings, *Whippomorpha* exhibited a high proportion of neuropeptide prohormone genes changing in the A, C, and V parameters that was similarly represented in *Mysticeti* and *Odontoceti* families.

The majority of neuropeptide prohormone genes exhibited changes between *Cetartiodactyla* suborders, while 18% of neuropeptide prohormone genes including promelanin concentrating hormone (PMCH), growth hormone releasing hormone (GHRH), PROK2, AVP, platelet derived growth factor subunit B (PDGFB), and pancreatic polypeptide (PPY) exhibited no substantial change in parameters. The conservation of the previous prohormone sequences supported amino acid substitution rates that are consistent with the JTT substitution model.

Among the neuropeptide prohormone genes that presented parameter changes across species, most changes occurred in one (31%) or two parameters (34%). These changes were primarily negative estimates for parameters A and V and positive estimates for parameter C. For some neuropeptide prohormone genes, including hepcidin antimicrobial peptide (HAMP), insulinlike 3 (INSL3), motilin (MLB), and spexin hormone (SPX), the amino acid parameter changes were observed in most taxonomic groups.

A notable finding stemming from the parameter coefficient estimates implies that changes in prohormone sequence lower the likelihood of large and hydrophobic amino acids (i.e., phenylalanine, tryptophan and tyrosine). These amino acids may compromise the neuropeptide function and therefore the evolutionary forces favor other types of amino acids. Moreover, our results indicate that the cation-π interaction provided by the aromatic amino acids, a strong noncovalent binding interaction that is important in protein secondary structure and interactions with drugs and neurotransmitters such as serotonin [66], is generally undesirable in prohormone sequences.

### 4.3. Domestication

The study of changes in neuropeptide gene protein sequences associated with domestication accounted for factors such as the distribution of domestic and wild species. Inference was partitioned orthogonally from taxonomy to remove all confounding groups while addressing differences in the number of species and limited gene flow. The evolution model indicated differences with wild terrestrial and domestic groupings. The domesticated *Ruminantia* had a slightly higher proportion of changes than wild *Ruminantia* with 60% of the neuropeptide prohormone genes being equal or had at least one parameter in common between domesticated and wild *Ruminantia*. This finding suggests that part of the differences between parameter values can be attributed to taxonomic differences rather than domestication.

Further challenges to the assessment of the association between domestication and neuropeptide gene changes stem from the domestication classification of some species and the genome assembly quality of other species. With respect to domestication assignments, for example, the gayl (*Bos frontalis*) is often considered the domestic form of the wild gaur (*Bos gaurus*) [67], however the degree of domestication is highly variable among the former group. With regards to sequence quality, the virtually identical nucleotide or protein sequences predicted from all camelid genomes available in this study invalidates the conclusion of adaptive introgression of endothelin 3 (EDN3) in South American camelids [6]. The association between domestication and prohormone sequence changes is further obscured by the counteraction of ancestral hybridization and artificial selection [3,4,6,68]. Considering the distribution of species across taxa and domestication groups and adjusting for variable assembly quality within and across domesticated groups, the present study identified notable and consistent changes among neuropeptide prohormone genes associated with docile and herdlike behaviors.

The neuropeptide prohormone genes that presented differences in amino acid parameters between domestic and wild species produce neuropeptides that participate in a vast array of functions. Among the neuropeptides from neuropeptide prohormone genes that have distinct sequence across species, APLN, AVP, TOR2X (salusin-containing torsin family 2 member A isoform), and VGF modulate angiogenesis, vasodilation, and vasoconstriction. Likewise, neuropeptides from ADCYAP1, CCK, GHRH, and VGF are associated with feeding and energy homeostasis. VGF is also associated with circadian rhythm, pain, memory, and learning [69,70]. In the context of domestication, a commonality is that the previous functions are associated with the sympatho-adrenomedullary system that involves the ‘fight or flight’ response where blood pressure and glucose levels increase in response to stress [71].

## 5. Conclusions

A study of the association between evolutionary and domestication processes and changes in neuropeptide gene sequences was undertaken. A bioinformatics pipeline was developed to identify the prohormone complement of 98 sequences across 118 *Cetartiodactyla* wild and domesticated species distributed across 4 suborders. An exhaustive survey of prohormone sequences was compiled, the sequences were compared, and evolutionary models were used to assess the change in amino acid properties including aromaticity, composition, polarity, and volume parameters. Remarkable findings include sequence differences among *Cetartiodactyla* that disrupt cleavage motifs in particular neuropeptide prohormone genes (e.g., INSL6, GNRH2, PRLH, POMC, CALC), potentially compromising the ability of the prohormone to generate bioactive neuropeptides. Similarly notable, the parameter coefficient estimates suggest that the evolutionary process tends to disfavor large and hydrophobic amino acids in neuropeptide prohormone genes. Evolutionary modeling indicated that some neuropeptide prohormone genes associated with the sympatho-adrenomedullary system and the ‘fight or flight’ response may be impacted by domestication. The prohormone complement provides the foundation for neuropeptidomic studies of medically and economically important characteristics and *Cetartiodactyla* species.

## Figures and Tables

**Figure 1 vetsci-09-00247-f001:**
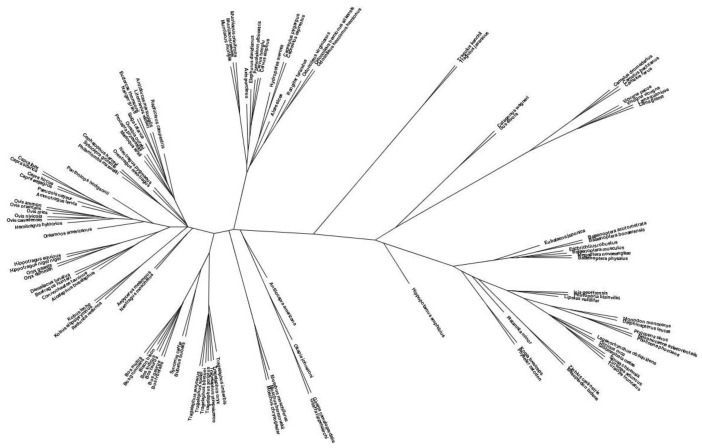
Species tree derived from individual prohomone gene trees.

**Table 1 vetsci-09-00247-t001:** Summary of domestic and wild *Cetartiodactyla* species used by taxonomic level.

Suborder and IF or Fam ^1^		N ^2^		Parvorder/Subfamily/Tribe/Genus ^3^
		D	W	
*Ruminantia*				
	*Bovidae*	7	50	*Bovinae*: 4,12; *Caprinae*: 2,12; *Alcelaphinae*: 0,4; *Reduncinae*: 0,3; *Cephalophinae*:0,3; *Hippotraginae*: 0,4; *Antilopinae*: 0,12; *Aepycerotinae*: 0,1
	*Cervidae*	1	15	*Muntiacinae*: 0,3; *Cervinae*: 0,5; *Hydropotinae*: 0,1; *Odocoileinae*: 1,6
	Other	0	9	*Moschidae*: 0,3; *Tragulidae*: 0,2; *Giraffidae*: 0,3; *Antilocapridae*: 0,1
*Suina*				
		1	o	*Sus*: 1,0; *Catagonus*: 0,1
*Tylopoda*				
	*Camelidae*	4	3	*Camelini*: 2,1; *Lamini*: 2,2
*Whippomorpha*				
		0	27	*Hippopotamidae*: 0,1; *Odontoceti*: 0,19; *Mysticeti*: 0,7
Total		13	105	

^1^*Cetartiodactyla* suborder and infraorder (IF) or family (Fam), ^2^ number of domesticated (D) and wild (W) species within infraorder or family, ^3^ the number of domesticated species followed by the number of wild species within each parvorder, subfamily, or tribe.

**Table 2 vetsci-09-00247-t002:** Number of neuropeptide prohormone genes by parameter specification in the evolutionary model.

Taxonomic Group ^1^	mS ^2^	Parameter ^3^							
		A		C		P		V	
		−	+	−	+	−	+	−	+
Overall									
All	81.5	52	5	4	64	17	19	51	3
Domestic	10	31	5	2	32	11	13	26	5
Wild	75.5	50	6	3	62	17	18	49	6
Wild terrestrial	56	51	3	1	56	11	23	48	3
*Ruminantia*									
All	55	51	5	3	50	13	24	51	5
Domestic	6	23	5	3	27	5	12	18	4
Wild	52	43	5	0	54	11	22	47	4
*Bovidae*	38	35	5	6	44	10	17	38	5
*Bovidae Antilopinae*	11	18	8	2	24	7	4	24	3
*Bovidae Bovinae*	10	22	1	6	20	10	6	15	6
*Bovidae Caprinae*	7	13	7	4	22	8	14	25	2
*Cervidae*	11	11	2	6	2	7	1	4	0
*Tylopoda*									
All	4	20	1	13	10	9	7	10	1
*Whippomorpha*									
All	20	32	3	6	23	13	11	39	0
*Cetacea Mysticeti*	5	16	6	4	16	18	8	13	2
*Cetacea Odontoceti*	14	30	3	7	21	11	10	31	3

^1^ Taxonomic group: All = All species overall or within each grouping, ^2^ Median number of species that have prohormone sequences across neuropeptide prohormone genes, ^3^ Number of neuropeptide prohormone genes with a negative (−) or positive (+) estimate in aromaticity (A), composition (C), polarity (P), or volume (V) parameter. Neuropeptide prohormone genes may have more than 1 modified parameter.

**Table 3 vetsci-09-00247-t003:** Parameter estimates for neuropeptide prohormone genes that presented changes in domestic or wild species.

	Parameter ^1^							
	A		C		P		V	
Symbol ^2^	All	Rum	All	Rum	All	Rum	All	Rum
Domestic								
APLN	3.62	9.80	0.00	0.00	0.00	0.00	0.00	0.00
Wild terrestrial								
ADCYAP1	−1.04	−1.27	0.00	0.00	0.00	−0.95	0.00	0.00
AVP	0.00	0.00	0.74	1.22	0.00	0.00	0.00	0.00
CCK	0.00	1.70	2.03	2.18	0.00	0.00	0.00	−0.90
GHRH	−1.35	−1.26	0.00	0.00	0.00	0.00	0.00	0.00
TOR2X	0.00	0.00	0.00	0.00	0.00	0.00	−1.32	−1.57
VGF	0.00	0.00	0.00	0.00	0.00	0.00	−1.23	−1.39

^1^ Estimated parameter change in individual aromaticity (A), composition (C), polarity (P), and volume (V) parameters within all domestic or wild species (All) and within domestic or wild *Ruminantia* species (Rum). ^2^ Neuropeptide prohormone gene symbol. APLN: apelin; ADCYAP1: adenylate cyclase activating polypeptide 1; AVP: arginine vasopressin; CCK: cholecystokinin; GHRH: growth hormone releasing hormone; TOR2X: torsin family 2 member A (salusin-containing isoform); VGF: VGF nerve growth factor inducible.

## Data Availability

Publicly available datasets were analyzed in this study. This data can be obtained from the National Center for Biotechnology Information.

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
