# Peer review of "Changes in Neuropeptide Prohormone Genes among Cetartiodactyla Livestock and Wild Species Associated with Evolution and Domestication"

_vetsci, 2022, doi:10.3390/vetsci9050247_

Round 1

Reviewer 1 Report

Basically, the authors compared neuropeptide prohormone genes among Cetartiodactyla livestock and wild species and indicated the evolution of these genes could be related to domestication. There are a few major shortcomings that prevent me from recommending publication of the current draft.

Major comments:

1) It’s unkown from this study that if these neuropeptide prohormones have undergone selection or accelerated evolution because no suitable analyses were done.  The authors may apply dn/ds analysis using PAML4 or other similar methods or softwares.

Yang Z. PAML 4: phylogenetic analysis by maximum likelihood. Mol Biol Evol. 2007 Aug;24(8):1586-91. doi: 10.1093/molbev/msm088.

2) It's also unclear what characteristic changes occurred in the phylogenetic tree.  Any changes should be mapped onto the tree and detailed explanations should also be made.

3) The effect of protein changes should be evaluated using more suitable softwares such as SIFT or PROVEAN:

Sim NL, Kumar P, Hu J, Henikoff S, Schneider G, Ng PC. SIFT web server: predicting effects of amino acid substitutions on proteins. Nucleic Acids Res. 2012 Jul;40(Web Server issue):W452-7.

Sandell L, Sharp NP. Fitness Effects of Mutations: An Assessment of PROVEAN Predictions Using Mutation Accumulation Data. Genome Biol Evol. 2022 Jan 4;14(1):evac004. 

Author Response

We appreciate for the opportunity to revise our manuscript VetSci-1644605. We have modified the submission to address questions and integrate the recommendations offered to us. The suggestions have strengthened our manuscript, and we thank the reviewers for their feedback.

Major comments:

1) It’s unkown from this study that if these neuropeptide prohormones have undergone selection or accelerated evolution because no suitable analyses were done.  The authors may apply dn/ds analysis using PAML4 or other similar methods or softwares.

Yang Z. PAML 4: phylogenetic analysis by maximum likelihood. Mol Biol Evol. 2007 Aug;24(8):1586-91. doi: 10.1093/molbev/msm088.

Reply: We agree that dn/ds analysis can be useful in particular conditions. However, nucleotide based methods such as the dn/ds approach were not applied because the present study favored the evaluation of changes that may impact the function of the gene product. Therefore the features corresponding to the protein sequence were studied. Also, understanding the changes at the amino acid level is necessary for the experimental detection and aligned with the statements in the abstract about our findings broadening the foundation for empirical studies about the function of the neuropeptidome associated with health, behavior, and food production.

2) It's also unclear what characteristic changes occurred in the phylogenetic tree.  Any changes should be mapped onto the tree and detailed explanations should also be made.

Reply: We agree on the value of mapping single protein changes. However, we must note that the phylogenetic species tree is different from the individual phylogenetic gene trees. In the context of our omic analysis, the visualization of the simultaneous mapping of all the differences from the 98 gene trees onto the species tree does not conduce the interpretation of the results.

3) The effect of protein changes should be evaluated using more suitable softwares such as SIFT or PROVEAN:

Sim NL, Kumar P, Hu J, Henikoff S, Schneider G, Ng PC. SIFT web server: predicting effects of amino acid substitutions on proteins. Nucleic Acids Res. 2012 Jul;40(Web Server issue):W452-7.

Sandell L, Sharp NP. Fitness Effects of Mutations: An Assessment of PROVEAN Predictions Using Mutation Accumulation Data. Genome Biol Evol. 2022 Jan 4;14(1):evac004. 

Reply: While SIFT and PROVEAN may be suitable in other contexts, these tools are inadequate for our data and goals because they rely on sequence conservation and ignore special features like cleavage sites. This is easily demonstrated by modifying the most common dibasic KR cleavage site to the next most common dibasic RR cleavage site. SIFT falsely reports a change protein function and PROVEAN a deleterious effect (-5.68 score). However, there is extensive experimental evidence that these sites are frequently cleaved such that the change would have no influence.

Reviewer 2 Report

Summary: 

The authors have carried out a study to infer neuropeptide genes using bioinformatic analysis. For this study they have created a large database of wild and domesticated species. Using sequence comparison including evolutionary models, authors find amino acid changes in neuropeptides to infer the basis of domestication. Their major findings include- changes in neuropeptide prohormone in Cetartiodactyla. Their findings suggest that hydrophobic amino acids were not favored evolutionarily and some neuro peptides have undergone changes which influence flight and fight response there by domestication. The manuscript is well written and the results were presented clearly. With the suggestions provided below, the manuscript could be improved.  

Minor
77-79: The sentence seems vague and misleading, genomes and transcriptomes can reveal multiple evolutionary patterns. RNAseq helps to differentiate expression and regulatory networks while selection scans from genomic data is useful.
129: Please specify what kind of filtering was disabled
134-135: apparent sequence duplication is not from assembly quality - absence of duplication / collapse of sequence is. Please rephrase
171: Please mention total neuropeptide prohormone genes in braces for relevance
175-179: Divergence from ancestral protein sequence can also cause problems in gene identification
213: Change “tended to have” into “had”
277: Remove “virtually”
278: 45 doesn’t seem like a relevant citation here
292: Please provide more details. What is this signal region, is this exonic or intronic, is this insertion previously reported and/or the evolutionary path known
299: “to all Ruminantia species” change to “in all Ruminantia species”
300-302: Pseudogenisation is inferred when pre-mature stop is present, please rephrase the sentence and mention why the assumption of pseudo-gene is valid here. Fragmentation is valid, but lacking complete prediction or indeterminate matches are not supporting evidences

Major
181-189: The species tree is based on one single gene family, this could be problematic if the gene family is known to undergo drastic changes within these families / species. Since the constructed species tree is congruent with the known existing species tree, please mention this in the results in addition to the already mentioned discussion. I would highly recommend constructing a species tree with single copy orthologs which can be obtained from BUSCO tool for higher confidence since older trees are based on fossils or limited markers
Figure 1: Please avoid overlapping tip labels and add support for the branching nodes
292: Add visualization for alignment
310-327: Add supplementary figures for visualization
1. Visualizations can be added to mention phylogenetic changes through illustrations of actual data representation. These have been given in wordy descriptions, visualization helps a lot more
2. Line 308 mentions RNAseq data. It would help to add perspective with a table regarding which species have RNAseq data available
3. Add a table on genome assembly qualities - N50, BUSCO scores definitely help to gain an understanding of the data used in the analyses
4. The genome assemblies are fragmented and sometimes poor quality, this necessarily doesn’t mean the gene is actually lost. The RNAseq data needs to be searched to confirm actual gene loss. RNAseq aligners accommodating sequence divergence can help in this analysis
5. Use protein to genome aligners such as SPALN (accounting for divergence) to make sure that the gene is actually absent and not a problem with tool sensitivity
6. It would be worth checking if these is any trend of positive selection or relaxation in wild vs domesticated species for these genes. Additionally, testing for evolutionary rate changes (using tools such as binary classification in RERconverge or Bayesian PhyloAcc) can help infer if there are differences in the two types

Author Response

We appreciate for the opportunity to revise our manuscript VetSci-1644605. We have modified the submission to address questions and integrate the recommendations offered to us. The suggestions have strengthened our manuscript, and we thank the reviewers for their feedback.

Minor
77-79: The sentence seems vague and misleading, genomes and transcriptomes can reveal multiple evolutionary patterns. RNAseq helps to differentiate expression and regulatory networks while selection scans from genomic data is useful.

Reply. We agree about the regulatory information that can be obtained from RNAseq experiments. However, as explained in subsequence sentence in our manuscript, the neuropeptide prohormone protein sequence is required to determine the consequences of evolutionary change.

129: Please specify what kind of filtering was disabled

Reply: Agree. We clarified that it was low-complexity filtering.

134-135: apparent sequence duplication is not from assembly quality - absence of duplication / collapse of sequence is. Please rephrase.

Reply: Throughout our work with prohormone sequences and assemblies (i.e., Amare et al., 2006; Delfino et al., 2010; Xie et al., 2010; Hu et al., 2016; Southey et al., 2020), and in agreement with the present study, we have frequently observed duplications  associated with prohormone sequences. However, the size of the duplications in this and previous studies suggest that most duplications are artifacts of the assembly process where contigs are incorrectly trimmed or joined. Rarely we have seen larger sequence duplications and these have not involve the complete gene. In consideration that the evidence suggest weaknesses in individual assemblies for some prohormones, rather than duplications across multiple species, we favor the present interpretation.

171: Please mention total neuropeptide prohormone genes in braces for relevance.

Reply: There are 98 genes and therefore the total number of genes is only 1 less than the percentage. In consideration of the high redundancy of both quantities, we favor the present description.

175-179: Divergence from ancestral protein sequence can also cause problems in gene identification

Reply: We agree that for certain sequences and studies divergence can cause problems. However, this was not a challenge in our study because we used many closely related species. The relationship between sequences enabled us to achieve robust alignments and gene identification.

213: Change “tended to have” into “had”

Reply: Agree, the sentence was edited accordingly.

277: Remove “virtually”

Reply: In the present study, the trees are not identical because the number of species used were different. Therefore, we favor the use of the adjective to note the high but not perfect consistency.

278: 45 doesn’t seem like a relevant citation here

Reply. Agree. The reference was removed.

292: Please provide more details. What is this signal region, is this exonic or intronic, is this insertion previously reported and/or the evolutionary path known

Reply. Agree. We replaced signal region with signal peptide. To our knowledge we are the first to highlight this insertion but currently there is limited experimental information to document our innovative finding.

299: “to all Ruminantia species” change to “in all Ruminantia species”

Reply. We note that the context of the sentence necessitates the use of the “to” rather than “in”.

300-302: Pseudogenisation is inferred when pre-mature stop is present, please rephrase the sentence and mention why the assumption of pseudo-gene is valid here. Fragmentation is valid, but lacking complete prediction or indeterminate matches are not supporting evidences

Reply: We note that from the established scientific definitions, a pseudogene is a gene that is considered to be nonfunctional. When a prohormone gene cannot provide bioactive neuropeptides then the gene is nonfunctional and, thus, the sequence fits the definition of pseudogene. Also, we provided evidence from other studies that support this conclusion.

Major
181-189: The species tree is based on one single gene family, this could be problematic if the gene family is known to undergo drastic changes within these families / species. Since the constructed species tree is congruent with the known existing species tree, please mention this in the results in addition to the already mentioned discussion. I would highly recommend constructing a species tree with single copy orthologs which can be obtained from BUSCO tool for higher confidence since older trees are based on fossils or limited markers

Reply: We note that the species tree is not from a single gene family as suggested in the reviewer’s statement. The species tree is derived 35 diverse gene families. The underlying algorithms to create species tree do address changes occurring within families. This was observed from the differences between the species tree and the individual gene trees discussed in the manuscript.

The comparison of the gene tree to the existing phylogenetic trees constitutes discussion and, therefore, this section was not included as a result of this work. Also, the species tree was constructed by “single copy orthologs” because almost all genes can be found in fish (as we reported in Hu et al., 2006).

Figure 1: Please avoid overlapping tip labels and add support for the branching nodes

Reply: Due to the number of species studied and the size of the page within the margins set by the journal, it was not feasible to create a tree without overlapping tip labels. Species trees including information from multiple genes lack measures of branch support that are found in single gene trees.

292: Add visualization for alignment

Reply: The visualization would not contribute insight into the functional differences between the sequences, nor add insights not already provided by other figures. Also, evidence from EST or long sequence RNA-seq data is required to provide context to the conserved and not conserved segments.

310-327: Add supplementary figures for visualization

Reply: The thousands of figures (~7400 were required) associated with the prohormones including all species, domestication status and taxonomic groups with all sequences or just the unique sequence were used. While necessary to the methodology, these provide no additional insights beyond the discussion. However, alignments with all species for each prohormone were added as supplementary material.

  1. Visualizations can be added to mention phylogenetic changes through illustrations of actual data representation. These have been given in wordy descriptions, visualization helps a lot more

Reply: Scientific editorial guidelines indicate the expectation that figures are explained and discussed within the manuscript. Therefore, adding figures would not remove the need for an adequate description of the changes

  1. Line 308 mentions RNAseq data. It would help to add perspective with a table regarding which species have RNAseq data available

Reply: We agree that relevant RNA-seq data can contribute insights. However, in the present study, the vast majority of the RNA-seq data is limited to mainly 5 domesticated species (cow, pig, sheep, goat, and dromedary camel). Also, most of this experimental data was not collected from central nervous system regions and therefore the relevance is limited.

  1. Add a table on genome assembly qualities - N50, BUSCO scores definitely help to gain an understanding of the data used in the analyses

Reply: The NCBI assemblies studied are listed in the manuscript and the N50 and other statistics are provided in the listed NCBI links and associated publications. These metrics are not extensively used to interpret results and the additions would advance the understanding of our findings. Lastly, the N50 and BUSCO scores are not relevant to the accuracy of neuropeptide prohormone gene prediction.

  1. The genome assemblies are fragmented and sometimes poor quality, this necessarily doesn’t mean the gene is actually lost. The RNAseq data needs to be searched to confirm actual gene loss. RNAseq aligners accommodating sequence divergence can help in this analysis

Reply: We agree that for all genes, extensive examination of all available data including different species and larger taxa groups was undertaken to assess the predictions. For vast majority of species studied, RNA-seq data is unavailable. The findings from our study may motivate research groups and funding agencies to dedicate resources for the collection of such information.

  1. Use protein to genome aligners such as SPALN (accounting for divergence) to make sure that the gene is actually absent and not a problem with tool sensitivity

Reply: We aree and our conclusion about gene loss was based on detection across multiple taxomic groups and multiple other independent studies. Thus, we are confident that our findings are not tool-dependent or a consequence of data problem.

  1. It would be worth checking if these is any trend of positive selection or relaxation in wild vs domesticated species for these genes. Additionally, testing for evolutionary rate changes (using tools such as binary classification in RERconverge or Bayesian PhyloAcc) can help infer if there are differences in the two types

Reply: We agree on the value of considering rates in other contexts. We did not investigate tests for selection or rate changes based on nucleotide data because neuropeptides result from cleavage of large proteins. A more comprehensive approach needs to include other domesticated species including cats, dogs, donkeys and horses.

Reviewer 3 Report

  • Line 53: add a reference
  • Line 100: add a phylogenetic tree showing the evolution of four Cetartiodactyla suborders
  • Line 159: add in the supplementary materials the list and sequence of all 98 prohormone protein
  • Paragraph Prohormone complement: add some information related to the function of cortistatin, RLN1, PYY2, islet amyloid polypeptide (IAPP) and galanin like peptide (GALP), TAC4, insulin-like 6 (INSL6), gonadotropin releasing hormone 2 (GNRH2), and prolactin releasing hormone (PRLH), POMC
  • Paragraph Evolutionary model: add some information related to the function of pro-melanin concentrating hormone (PMCH), growth hormone releasing hormone (GHRH), PROK2, AVP, platelet derived growth factor subunit B (PDGFB), and pancreatic polypeptide (PPY), hepcidin antimicrobial peptide (HAMP), insulin like 3 (INSL3), motilin (MLB), spexin hormone (SPX)
  • References: follow author’s guideline https://www.mdpi.com/journal/vetsci/instructions : doi is not required, add pp. before page number

Author Response

We appreciate for the opportunity to revise our manuscript VetSci-1644605. We have modified the submission to address questions and integrate the recommendations offered to us. The suggestions have strengthened our manuscript, and we thank the reviewers for their feedback.

Comments and Suggestions for Authors

  • Line 53: add a reference

Reply: Agree. A reference was added.

  • Line 100: add a phylogenetic tree showing the evolution of four Cetartiodactyla suborders

Reply: The proposed tree would be redundant to other tables and figures already in the manuscript and we favor the present results that also convey information on other species.

  • Line 159: add in the supplementary materials the list and sequence of all 98 prohormone protein

Reply: Agree. A link to the data was provided as supplementary data.

  • Paragraph Prohormone complement: add some information related to the function of cortistatin, RLN1, PYY2, islet amyloid polypeptide (IAPP) and galanin like peptide (GALP), TAC4, insulin-like 6 (INSL6), gonadotropin releasing hormone 2 (GNRH2), and prolactin releasing hormone (PRLH), POMC

Reply: These prohormones have not been widely studied in Cetartiodactylal species so there is no information for most species studied. On the other hand, in human and rodents, neuropeptides prohormones modulate multiple biological processes and the review of the known functions on a handful of species would detract from the manuscript focus.

  • Paragraph Evolutionary model: add some information related to the function of pro-melanin concentrating hormone (PMCH), growth hormone releasing hormone (GHRH), PROK2, AVP, platelet derived growth factor subunit B (PDGFB), and pancreatic polypeptide (PPY), hepcidin antimicrobial peptide (HAMP), insulin like 3 (INSL3), motilin (MLB), spexin hormone (SPX)

Reply: Similar to the previous set, this set of prohormones has not been widely studied in Cetartiodactylal species so there is no information for most species studied. On the other hand, in human and rodents, neuropeptides prohormones modulate multiple biological processes and the review of the known functions on a handful of species would detract from the manuscript focus.

  • References: follow author’s guideline https://www.mdpi.com/journal/vetsci/instructions : doi is not required, add pp. before page number

Reply. We note that the author instructions refer to the MDPI Reference List and Citations Guide (https://www.mdpi.com/authors/references). These guides states that “Journal references must cite the full title of the paper, page range or article number, and digital object identifier (DOI) where available” (emphasis added). These guides are also provided in the reference styles section.

Round 2

Reviewer 1 Report

Almost none of my and other reviewers' major comments were properly addressed, only minor changes were done.  There are still major shortcomings that prevent me from recommending the publication of the current draft.

Author Response

We disagree as Reviewer 1 made no requested changes and the suggestions were not appropriate for neuropeptide prohormone genes or studying domestication in general.

  • Reviewer 1 only suggested a standard approach to detect selection at the nucleotide level. We pointed out that this is inappropriate for neuropepeptide prohormone genes because changes must be evaluated at the protein level. It should have been obvious to Reviewer 1 that this approach cannot be used to study domestication due to multiple reasons. Since species were domesticated at different times and often vastly geographic locations and environments, it is unlikely that exact same changes would be present or even detectable in all domesticated species. This is further complicated by multiple domestication events have been documented within pigs, cattle and South American camelids. Natural selection is completely confounded with selection involving domestication because the dn/ds method uses changes since species diverged from the last common ancestor instead of changes when the species were domesticated. This concern does not change our results or conclusions due to our species level comparison.
  • Reviewer 1 suggested mapping “characteristic changes”. No changes were needed because there were no “characteristic changes” that would apply to the provided gene tree.
  • Reviewer 1 suggested using different software to evaluate protein sequences. As we pointed out that these software completely fails to account for the features of neuropeptide prohormone proteins. We provided what is the most simplistic change (cleavage sites) to neuropeptide proteins that SIFT and PROVEAN completely failed to predict.

Reviewer 3 Report

thanks for replying to my comments

Author Response

Thank you very much as your contributions enhanced our manuscript.